# A Smart Rig for Calibration of Gas Sensor Nodes

**DOI:** 10.3390/s20082341

**Published:** 2020-04-20

**Authors:** Mohieddine A. Benammar, Sabbir H. M. Ahmad, Abderrazak Abdaoui, Hasan Tariq, Farid Touati, Mohammed Al-Hitmi, Damiano Crescini

**Affiliations:** 1Electrical Engineering, College of Engineering, Qatar University, P.O. Box 2713, Doha, Qatar; mbenammar@qu.edu.qa (M.A.B.); sabbir@qu.edu.qa (S.H.M.A.); hasan.tariq@qu.edu.qa (H.T.); touatif@qu.edu.qa (F.T.); m.a.alhitmi@qu.edu.qa (M.A.-H.); 2Department of Information Engineering, Brescia University, 25121 Brescia, Italy; damiano.crescini@unibs.it

**Keywords:** gas sensors calibration, cross-sensitivity, piecewise curve fitting, IEEE 802.15.4 (ZigBee), linear regression, mass flow controller, wireless calibration

## Abstract

Electrochemical gas sensors require regular maintenance to check and secure proper functioning. Standard procedures usually involve testing and recalibration of the sensors, for which working environments are needed. Periodic calibration is therefore necessary to ensure reliable and accurate measurements. This paper proposes a dedicated smart calibration rig with a set of novel features enabling simultaneous calibration of multiple sensors. The proposed calibration rig system comprises a gas mixing system, temperature control system, a test chamber, and a process-control PC that controls all calibration phases. The calibration process is automated by a LabVIEW-based platform that controls the calibration environment for the sensor nodes, logs sensor data, and best fit equation based on interpolation for every sensor on the node and uploads it to the sensor node for next deployments. The communication between the PC and the sensor nodes is performed using the same IEEE 802.15.4 (ZigBee) protocol that the nodes also use in field deployment for air quality measurement. The results presented demonstrate the effectiveness of the sensors calibration rig.

## 1. Introduction

Gas sensors are used in various monitoring of air quality and control applications. Some types of these sensors are used for monitoring and controlling the combustion processes [1,2,3]. Oxygen and other types of sensors are used in medical applications, for example, for analyzing human breath [4,5,6]. Gas sensors may also be used in electronic olfaction schemes for odor detection and identification [7,8,9,10]. Gas sensors may even be used to evaluate fruit ripening [11,12]. Different types of gas sensors are used for detecting hazardous gases in mines, petrochemical installations, and other applications [13,14,15,16]. Gas sensors are evidently widely used for the monitoring of air quality for indoor and outdoor environments.

Air-quality monitoring is extremely important as it has a direct impact on human health. The attention toward the risks that are related to exposure to chemical pollutants is rapidly increasing because of the well-established correlation between these pollutants and human diseases. Some studies link millions of death each year to air pollution. The air quality, for both indoor and outdoor, is important to comfort levels and human health, and thus requires continuous monitoring. In recent years, indoor air quality has received more attention and research than outdoor air quality, due to the fact that (i) the concentration of some pollutants is two to five times higher indoor than outdoor and (ii) people, and in particular the elderly, spend up to 90% of their time indoors [17].

Air quality has been traditionally monitored using networks of routine static measuring stations often supplemented by modeling [13,18,19,20]. These stations are usually reliable and can accurately measure a wide range of air pollutants using traditional analytical instruments, such as mass spectrometers and gas chromatographs. Each unit is indeed a set of expensive chemical analyzers that are very similar to those employed in a chemical laboratory. This is reflected in the cost of routine measuring instruments resulting in a sparse distribution of stations and consequently, a relatively coarse spatial resolution in air pollution monitoring. The spatial resolution of the resulting pollution maps is inadequate for identifying pollution hot-spots and in accounting for the temporal and spatial variability of air pollution on urban scales, especially when aiming at human exposure assessment [21,22,23].

In order to achieve the required high-density spatiotemporal resolution, sensor networks employing large numbers of low-cost sensors is a viable option. However, low-cost gas sensors, especially those based on electrochemical principles, suffer from temperature and humidity effects, long-term drift, aging problems, and cross-sensitivity. Hence, frequent calibration of such devices is usually required to guarantee reliability of measurements. To the best of our knowledge, the prices of some low-cost electrochemical sensors are around tens of United States Dollar (USD). In addition, the calibration of other nonelectrochemical sensors—such as the self-actuating and self-sensing gravimetric ethanol sensor based on 3D ZnO-Nanorods@Si-Nanopillars [24] and ethanol sensor based on efficient self heated GaN Nanorods [25] can be also performed using the calibration rig that we propose in this paper.

Note that our calibration system is applicable for any sensors. We used commercial sensors just as an application which is close to the real air-quality monitoring needs. In addition, commercial sensors are widely employed (Solidsense, Honeywell, Alphasense, Figaro). Calibration can be done in the field [26,27,28,29,30,31,32] or in environmental chambers under controlled conditions in the lab [33,34,35]. The former method relies on colocating the sensor to be calibrated with a standard and precise instrument against which calibration is performed, which would require long-term data collection and relies on the unpredictable range of change of the concentration of the target gas. On the other hand, laboratory calibration in environmental chambers can be done under user-controlled conditions. The process of gas sensor calibration is tedious and includes (i) collection of sensor response at different operating points, (ii) data processing and determination of an appropriate mathematical model linking the real gas concentration to the sensor response, and (iii) storing of the calibration model or look-up tables in a memory associated with the sensor to correct its readings during subsequent deployments. To make the low-cost gas sensor usable, it is important to increase its accuracy by developing a real-time adaptive calibration [36,37,38,39,40,41].

In [42], the authors propose an environment-adaptive calibration system dedicated to the implementation of a collaborative calibration technique for outdoor low-cost electrochemical gas sensors. The adaptive calibration system they propose uses a new practical calibration algorithm operating on the Raptor IoT cloud platform that was developed as part of the Horizon 2020 Captor project which is a collective awareness platform for tropospheric ozone pollution. Their platform—formed by 60 nodes, deployed in Italy, Spain, and Austria—consisted of 140 metal-oxide O_3_ sensors, 25 electrochemical O_3_ sensors, 25 electrochemical NO_2_ sensors, and 60 temperature and relative humidity sensors [43].

This paper presents an automated calibration rig for the simultaneous calibration of multiple sensors; the calibration steps are preprogrammed and the operator intervention is minimal. We employ LabVIEW to monitor the calibration and data measurements for the calibration rig. This LabVIEW-controlled rig enables simultaneous calibration of several wireless sensor nodes that include multiple sensors for multiple target gases in succession automatically. In addition, the test chamber is fitted with temperature control capability for testing the sensors at temperatures above the ambient. The chamber and the mixing pipes have a volume of about 1000 mL and the maximum achievable temperature is 60 °C.

A suitable mathematical algorithm, based on piecewise approximation, is developed for the calibration of each sensor under test. The rig is fitted with a dedicated communication protocol to communicate with the sensor board for the data collection phase and for uploading the calibration equations onto the sensor board. To the best knowledge of the authors, the approach of automatic calibration of sensor nodes has not been reported before. The remainder of the paper is organized as follows. Section 2 presents the indoor air quality gas sensing node investigated in the present work. Section 3 describes the hardware and software operation details of the calibration rig. Section 4 presents the results of the calibration rig with further discussions. Section 5 concludes the paper.

## 2. Indoor Air Quality (IAQ) Sensing Node

The proposed automated rig may be used to calibrate any gas sensors or gas sensor nodes. However, the presented calibration method is applied to wireless sensor nodes used for measurement of indoor and outdoor air quality and described in [44]. The hardware for the sensing node has been designed using sensors from Libelium. Based on their principle of operation, there are two groups of gas sensors used in the node—electrochemical sensors for CO, Cl_2_, O_3_, NO_2_, and SO_2_; and nondispersive infrared (NDIR) CO_2_ sensors, as shown in Table 1. The electrochemical sensors operate in the amperometric mode, i.e., they output current, which depends on the concentration of the target gas [45]. Electrochemical gas sensors are gas detectors that measure the concentration of a target gas by oxidizing or reducing the target gas at an electrode and measuring the resulting current. The sensors contain two or three electrodes, occasionally four, in contact with an electrolyte. The electrodes are designed by fixing a high-surface-area precious metal on to the porous hydrophobic membrane. The working electrode contacts both the electrolyte and the ambient air to be monitored. The electrolyte most commonly used is a mineral acid, but organic electrolytes are also used for some sensors. The electrodes and housing are usually in a plastic housing which contains a gas entry hole for the gas and electrical contacts [24].

The NDIR sensors [46] generate a voltage signal whose amplitude depends on the amount of infrared light of a specific wavelength absorbed, which in turn, depends on the target gas concentration. Each sensor is equipped with an Analog Front End (AFE) interface that includes the electronics that run the sensor and an EEPROM in which sensor-related information (e.g., sensor type, gas type, measurement range) and factory calibration data are stored. The AFE for each amperometric sensor (other than CO_2_) includes a transimpedance amplifier that converts the sensor current into voltage. Note that the sensors are factory-calibrated and that the calibration data is stored in the EEPROM of the AFE; however, sensor recalibration by the user cannot be performed by making use of the EEPROM. Hence, under the proposed calibration method described below, the calibration data is stored in a micro Secure Digital (SD) card integrated into the embedded board of the sensor node. The sensor-related information and sensor measurement data are available to the user through a digital I^2^C communication bus. Additional details about the Libelium sensors may be found in [44,47].

The key components of the sensor node hardware are illustrated in Figure 1. Each node incorporates two boards: a board on which sensors are plugged (left side) and an embedded processor board (right side). The latter board includes mainly a microcontroller, a ZigBee wireless communication module Xbee-PRO (which is a XBee Znet 2.5 OEM RF module from Digi), and a micro-SD card. The microcontroller processes the sensor data and handles ZigBee wireless communications. Additionally, the microcontroller stores sensor readings in a local SD card and translates these readings into gas concentrations by using respective calibration equations for every gas sensor on the node. The calibration equations are stored in the SD card after every recalibration.

## 3. Calibration Rig Design

The hardware of the automated-personal computer (PC) based test rig is composed of a gas blending system and a temperature control system. Calibration is done wirelessly in order to make the process seamless. Identical sensors are calibrated together under same conditions for consistency and accuracy. The calibration data for each sensor is used to generate piecewise functions (using a mathematical algorithm) whose coefficients are stored locally in the PC under an associated unique identifier (ID). The sensor node stores the calibration data in the micro-SD card for further air-quality monitoring. For recalibration, the sensor nodes are retreated from the field and placed in the rig where the existing calibration data gets overwritten with the latest calibration results.

### 3.1. Calibration Rig Hardware

#### 3.1.1. Gas Mixing System

Figure 2 shows a simplified diagram of the proposed setup for the calibration rig, which includes an array of gas cylinders, a gas blending system that controls mixture composition, a sealed chamber with temperature control system, and a process control PC unit. The gas blending is performed using mass flow controllers (MFCs). The composition of the test gas mixture resulting from blending injected to the sealed chamber is adjusted by controlling the MFCs using the control PC through flow of individual gas components. The MFCs have the standard RS485 communication interfaces to interface them to the PC that runs the rig. The outlet of the chamber is connected to a bubbler to seal the system and expel test gases safely into the atmosphere through appropriate piping. One of the gas cylinders contains pure air and the other contains premixed target gases (diluted in air, with fixed concentration $g_{X}^{c}$ in ppm). To set the concentration *g_X_* of the target gas X in the sealed environmental chamber, the flow rates *f_X_* and $f_{AIR}$ of premixed target gas and air, respectively, are adjusted by $MFC_{X}$ and of $MFC_{AIR}$ according to (Equation 1),
(1)$$f_{X}=\frac{g_{X}}{g_{X}^{c}}\left(f_{X}+f_{AIR}\right).$$

Evidently, the dilution of the target gas requires mixing it with at least another gas, for example, air, in which case $f_{total}=f_{X}+f_{AIR}$. Hence, either the flow of pure air ($f_{AIR}$ ) is fixed first and the flow of the diluted gas (*f_X_*) is determined, or the total flow ($f_{X}+f_{AIR}$) is fixed and both *f_X_* and $f_{AIR}$ are determined.

#### 3.1.2. Temperature Control System

At the heart of the temperature control system are two 24-voltsrated 100-W heaters. The heaters are controlled using two insulated-gate bipolar transistor (IGBT) driver modules that are controlled using an Arduino Uno Microcontroller. The temperature inside the chamber is measured using an LM35 temperature sensor from Texas Instruments (Texas USA). Besides that, two fans are installed to expedite convection inside the sealed test chamber for thermal homogeneity. The microcontroller receives the temperature set point from the process control PC then uses a closed-loop hysteresis control algorithm to control the temperature based on the reading of the temperature sensor. The unit can establish sealed chamber temperatures from room temperature up to 60 °C. Figure 3 presents the schematic of the temperature control system.

### 3.2. Computer and Software Tools

The LabVIEW platform has been used to develop the software that controls the entire calibration process. Through the developed software, the calibration rig PC achieves the two main following tasks:Control the composition of the gas and temperature inside the chamber.Calibration processing and storing the calibration data of the sensor node under test.

First, based on the test requirements, the software programs the MFC flow rates serially over RS485 communication link to control the desired gas composition in the chamber. For each test point, the software reads and stores the sensor data locally in the PC. The communication with the node is accomplished using IEEE 802.15.4 (ZigBee) communication interface. The graphical user interface (GUI) in Figure 4 shows the virtual instrument (VI) created in the LabVIEW environment for running the calibration rig and generating the sensor output as a function of the set point gas concentrations. In this GUI, the user selects the parameters of the experiment such as the Gas type, the concentration at the cylinder, the serial port of the Xbee Pro (sensor reading) and the serial port of the two flowmeters (gas and air), and finally the initial concentration and the concentration step of the gas and the time duration for each concentration (seconds). The results are displayed in the GUI given by Figure 5. The instantaneous values of the six reading, from the sensor board, are displayed in the upper-side. However, the curve in the bottom side shows the historical values of the measurements for one of the six sensors. The behavior of the curve enables us to detect any problem related to the experiments (gas leakage, overflow at one flowmeter etc...) during the experiment.

Secondly, the calibration of sensor nodes is accomplished in two steps, as shown in Figure 6.
Testing the node at various concentrations of each target gas, collecting the data, and generating a calibration equation for each sensor (identified by its unique ID which is stored in the EEPROM of its AFE).Uploading the calibration equations into the SD card of the sensor node.

There are two possible ways of calibrating the node sensors:Sensors of the same type are mounted on the same node and calibrated together. Calibrating identical sensors together provides two-fold benefits: it reduces the volume of gas required for calibrating sensors, and it ensures that all sensors have been calibrated for the same test points under same conditions.Multiple sensor nodes, with their associated various sensors, may be calibrated together. This approach enables direct assessment of sensors’ cross-sensitivity to all the gases being injected in the chamber.

Calibration is performed wirelessly over Xbee communication link, which is adopted for the wireless sensor nodes [44]. The nodes and the calibration rig have Xbee PRO series 2 ZigBee radio modules. These radio modules can be operated in two different modes, namely, API and AT mode (called transparent mode). For our purpose, the AT mode has been used primarily for its simplicity of implementation, low payload, and low latency. In addition, this mode works for two-way communication between ZigBee devices such as node to/from controller.

A dedicated application protocol based on master–slave communication structure has been designed, where the process control PC acts as the master and the sensor node as the slave. The protocol has been designed considering speed and reliability of communication, taking into account the simplicity of implementation. The protocol is based on byte level data packets rather than strings to improve the speed of communication. In addition, cyclic redundancy check (CRC) bytes are appended to check for message corruption during communication in transparent mode.

#### 3.2.1. Collection of Sensors Measurement Data

For the collection of data, sensor nodes are placed inside the environmental chamber of the calibration rig (Figure 6). During this process, the sensor node should be programmed to operate in measurement mode, which is the use of the sensor node after calibration. Through the sensor node firmware, when the measurement mode is selected, the (voltage/current) data collected from the sensors are directly transmitted through ZigBee to the LabVIEW PC. However, after calibration and uploading the polynomial coefficients in the SD card, the voltage measurements collected from the sensors are converted to concentration (in ppm) by applying the associated curve fitting identified by the serial number of the sensor. As discussed above, the MFCs connected to the PC are used to set specific target gas concentrations inside the chamber by mixing streams of target gases with pure air according to Equation (Equation 1). It is therefore critical to assess the time needed to wait before the gas concentration in the chamber settles down following a computer command to change the flow rates of the MFCs. Therefore, a test is needed to estimate the response time before acquiring the sensor reading under test. The wait time includes both the time needed to flush out the previous gas mixture and the response time of the sensor itself.

The calibration can be conducted by stepping up/down the gas concentration in the chamber and recording the sensor response over time, as shown in Figure 7.

Following the test mentioned above, the wait time can be determined for each sensor for various concentrations of the relevant target gas. For calibration purposes, the range of concentration of a given target gas X is determined according to the expected range in indoor air. Nevertheless, a minimum number of test points within this range is needed to ensure that any nonlinearity of sensor response within the calibration range will be recorded. As described previously, the rig allows simultaneous testing of multiple sensors/sensor nodes. The LabVIEW software starts by identifying the IDs (stored in the EEPROM of the AFE of the sensor) of all sensors placed in the environmental chamber of the test rig. These IDs include the type X of target gas, which will be used to sort out the various types of sensors for subsequent testings. A file is created for each sensor; the file name is composed of the sensor ID (including the absolute serial number) for easy identification. In addition, the identification of the sensor type leads to knowledge of the corresponding target gas; prestored set point concentrations are stored in the LabVIEW VI for each target gas X. The sensors may be tested at real-world operating temperatures. The temperature may be set in the environmental chamber using the temperature controller. However, since the deployment is for indoor air-quality monitoring, temperature is not expected to vary significantly as indoor temperature is usually adjusted for thermal comfort of people in these indoor environments. If, however, temperature is expected to vary significantly to a point that it does affect sensors’ outputs, the rig allows finding the correlation of sensors’ outputs with ambient temperature for ultimate temperature compensation of these sensors’ outputs. The data collection, for each set of sensors with target gas X, starts by setting first the concentration of target gas (from the prestored set points) through control of the corresponding MFCs according to Equation (Equation 1). Then, the program waits for the prestored wait time before reading the outputs of the set of sensors for target gas X. Averaging for multiple readings for each set-point ensures minimization of the effects of random noise. The set-point concentration (SPC) and the corresponding sensor output are then stored in the corresponding sensor file. The graphical user interface of the curve-fitting algorithm is used to read the measurement file of a set of sensors already generated during the first step of the calibration. Figure 8 illustrates the GUI of the curve fitting where we can see two main parts: calibration input and calibration results. In the calibration input, we just personalize the location of the file containing the calibration readings, the order of the polynomial, and the *R*^2^ value. The software starts by finding a linear fit for the calibration data with an *R*^2^ value higher than the required target value. If not possible, the software then raises the order of the equation until an equation that satisfies the target *R*^2^ value is obtained.

The LabVIEW module reads the file and automatically extracts the ID (absolute serial number) of all the gas sensors registered in the data file as a (set point concentration, voltage measurements) averaged on 5 values. The polynomial coefficients are produced for each sensor node and the results are displayed in the Calibration results part. The polynomial coefficients are then stored in a text form of the following form (Figure 9):

A short description of each parameter is given below. This polynomial fit operates by putting the arrays of X and Y values (sensor readings and set point concentrations, respectively). The weight is the array of weights for the observations (X, Y). If the weight is left disconnected, as in the present application, the polynomial fit sets all elements of weight to a default value equal to 1. The tolerance input determines when to stop the iterative adjustment of Polynomial Coefficients when either the Least Absolute Residual or the Bisquare method is used. For the first method, if the relative difference between residue in two successive iterations is less than the tolerance, this polynomial fit returns the resulting Polynomial Coefficients. For the second method, if any relative difference between Polynomial Coefficients in two successive iterations is less than the tolerance, this VI returns the resulting Polynomial Coefficients.

Note that, in our application, the input to the Standard Deviation, Variance VI is the array *Y* (see Figure 10), and the *w* in this VI is W the array of weights for the observations (X, Y). If *w* is not wired, the VI sets all elements of weight to 1. The total sum squared is defined as follows [48]:(2)$$SST=\frac{1}{N}\sum_{i=1}^{N}w_{i}{\left(y_{i}-\bar{y}\right)}^{2}.$$

The sum of squares error (SSE) is defined as follows :(3)$$SSE=\frac{1}{N}\sum_{i=1}^{N}w_{i}{\left(y_{i}-f\left(x_{i}\right)\right)}^{2}.$$

Figure 11 depicts details of the LabVIEW software algorithm used for the implementation of the curve fitting and the generation of the calibration polynomial fit for each sensor. The fitting procedure requires uploading the collected experimental data into the sample arrays *X* and *Y* of the LabVIEW VI. The experimental data is assumed to have been collected during step 1, depicted in Figure 6. The General Polynomial Fit VI described above (Figure 10) is then used to fit a polynomial to the experimental data. The coefficients of the polynomial together with the goodness of fit parameter(s) are provided by the General Polynomial Fit VI. There are various parameters that may be used to evaluate how good is the fitting curve to the data. Since the least square method is used for implementing the fitting algorithm, in the General Polynomial Fit VI, the optimization of the polynomial coefficients is done by minimizing the residue that represents the variance given in Equation (Equation 3). This optimization assumes that the polynom order (PO) is already set. This residue (sum of squares error, SSE) and the variance output of the Standard Deviation and Variance VI are used to determine *R*^2^ of the best fit using (Equation 4).
(4)$$R^{2}=1-\frac{SSE}{SST}$$
*R*^2^ is then used to assess whether the resulting PO is acceptable. If *R*^2^ is below a predetermined target, then the PO is incremented and the General Polynomial Fit VI runs again to find new polynomial coefficients and associated residue. The process is repeated until the value of *R*^2^ reaches the predetermined target or the PO reaches its highest permitted value Therefore, the maximum PO depends on the number of test points used for a given sensor. At the end of the curve-fitting process, the time-stamped polynomial coefficients and *R*^2^ value are stored on the sensor calibration file in the process control PC hard disk where the file is identified by the unique sensor ID.

#### 3.2.2. Sensor Calibration

As shown in Figure 6, the polynomial fitting, applied to the data collected from each sensor, results in finding the polynomial coefficients that are used for the calibration of the sensor. The process of uploading the sensor-specific polynomial coefficients to the sensor node is straightforward Figure 12. During this process, the sensors to be calibrated are placed in their respective nodes which should be programmed to operate in the calibration mode. The calibration process starts by reading the IDs of the sensors of the nodes and then, after checking, stores the calibration data to the relative sensor on the board. During monitoring mode, the ID of each sensor present on the sensor node is then used to identify the corresponding polynomial coefficients stored in SD card of the sensor board. Each measured voltage (in mV) indicating the gas concentration is converted to calibrated concentration (ppm) by applying the polynomial function of the identified sensor.

### 3.3. Modelization of the Calibration Drift

The conventional sensor calibration consists of relating the steady-state sensor responses to the target analyte concentrations [36,37]. Based on the experimental data collected following the proposed design of experiments strategies, transfer function (TF) models are estimated to calibrate the dynamic behavior of a sensor. Such dynamic calibration models enable the use of transient sensor signals to track the rapid change in the target analyte concentration.

#### Transfer Function Modeling

In the following, we denote by u(t) the sensor response usually in (mV), and c(t) the target analyte concentration in (ppm). The time *t* index is assumed to be discrete as t=1,2,3.... The task of dynamic calibration is to quantify c(t)−u(t) by a mathematical model extracted with the help of experimental data. Transfer function have been originally introduced in signal modeling for estimating damped sinusoids and exponentials [49,50]. In [37], the authors, for the first attempt, use TF models to fully calibrate the dynamic profile of a sensor, and do assist the real-time monitoring of evolving environments. With the static sensor calibration, forward and inverse models are respectively discussed.

**Forward transfer function model:** This model approximates the sensor response u(t) with respect to the historical sensor responses and the immediate previous analyte concentration c(t) [37]:(5)$$u\left(t\right)=F\left(c(t-1);u(t-1),u(t-2),\ldots ,u(t-L)\right).$$
The integer *L* denotes the time order of the model which is according to [51] and [37] limited to 3. As in [37] the TF in (Equation 5) is assumed to be polynomial with order *P*. A full quadratic model with *P* = 2 is given as :(6)$$\begin{array}{ll} u(t) & =b_{0}+\left(k_{1}c\left(t-1\right)+\sum_{l=1}^{L}b_{l}u\left(t-l\right)\right)+\left(k_{11}c^{2}\left(t-1\right)+\sum_{l=1}^{L}b_{ll}u^{2}\left(t-l\right)\right)\\ & +\left(\sum_{l=1}^{L}h_{l}c\left(t-1\right)u\left(t-l\right)+\sum_{i=1}^{L-1}\sum_{j=i+1}^{L}h_{ij}u\left(t-i\right)u\left(t-j\right)\right) \end{array}$$
where all the coefficients are unknown parameters. In general, order *P* is not greater than 3 [52].

**Inverse transfer function model:** The inverse transfer function model approximates time-varying analyte concentration in terms of the sensor responses:(7)$$c\left(t-1\right)=G\left(u(t),r(t-1),\ldots ,u(t-L)\right)$$
For inverse model, the authors in [37] adopt a general polynomial form with non linear (higher-order) terms which is consistent with possible nonlinearity involved in its forward counterpart
(8)$$c\left(t-1\right)=\sum_{l=1}^{L}\sum_{i=0}^{P}{\left[u(t-l)\right]}^{i}+\sum_{l_{1}=0}^{L-1}\sum_{l_{2}=l_{1}+1}^{L}\sum_{i=1}^{P-2}\sum_{j=i+1}^{P-1}h_{ijl_{1}l_{2}}{\left[u(t-l_{1})\right]}^{i}{\left[u(t-l_{2})\right]}^{j}$$
with *L* = 3, the inverse model can be approximated as:(9)$$c\left(t-1\right)=a_{0}+a_{1}u_{s}\left(t\right)+a_{2}u_{s}{\left(t\right)}^{2}+a_{3}u_{s}{\left(t-1\right)}^{2}$$
where $u_{s}\left(t\right)=\left(u(t-1)u(t)+r(t+1)\right)/3;\;t=1,2,N$. $u_{s}\left(t\right)$ is defined to give smoother behavior of u(t). We can see in the calculation of the polynom coefficients by LabView, that the order of the polynom will not exceed 2 which confirms the models given in (Equation 9).

## 4. Results

In this section, we present the implementation of the three calibration steps detailed in the previous sections for CO and CO_2_ sensors. First, we collect the measurement data for each sensor separately. Then, based on the set point concentrations for each gas and the measurement for each sensor node, we use LabVIEW to generate the polynomial fitting coefficients. Finally, we update the calibration file on the SD card for each sensor. At least one additional set of measurements is carried out to verify the results (see Figure 13 and Figure 14 below). In these calibration experiments, the concentration of each target gas was varied in steps. We found that the wait time depends on the gas flow rate used in the experiments, on the volume of the tubing, on the environmental chamber, on the number of sensor nodes inside the chamber, and evidently, on the response time of sensors.

Figure 13a shows the response time of three CO sensors before calibration that were exposed to the same CO concentration stepping. Although the sensors have the same specifications, they are not identical. Thus, the response voltages produce different calibration curves, as shown in Figure 13b. Obviously, the output is different for all sensors, and this can be accounted for by the inherent aging and drift effects of electrochemical sensors. However, for consistency, the outputs have to be the same under the same gas condition and hence, the sensors have to be calibrated. Figure 14 depicts the results we obtained after calibrating the same previous three CO sensors, where the sensors outputs (in mV) are converted to their ppm equivalents by using the coefficients of the fitting polynomials read from the SD card. The coefficients of the best polynomials that were obtained during the calibration process are shown on the inset.The data collection used for calibration was done at room temperature (22 °C), and the target *R*^2^ was set to 0.999. Recall that the target *R*^2^ represents the required minimum value for *R*^2^ of the polynomials, and therefore the target *R*^2^ was used by the VI to determine the polynomial fitting. It is obvious that the calibration has rigorously corrected the sensors discrepancies in terms of response and output level.

We repeated the above procedure for the CO_2_ sensor. Figure 15 shows the measurement results and the evaluation of the CO_2_ sensor before calibration. Figure 16 shows the test of the CO_2_ after calibration: the plots give the calibrated measurement (ppm) versus the set point concentration (ppm) (as in Figure 14). In addition, after calibration, all the three CO_2_ sensors, exposed to same gas concentrations, give identical results.

## 5. Conclusions

In this paper, we presented a new practical calibration rig system that is dedicated to gas sensors for outdoor and indoor air-quality monitoring. We proposed a complete calibration framework with a set of novel features. The calibration rig has been completely automated by developing a LabVIEW-based platform that controls the whole process such as controlling the calibration phases, ensuring data transmission/reception, and setting the flow rate of each mass flow control. The results demonstrate the effectiveness of the sensor calibration rig and the accuracy of the measurements of each calibrated sensor. In future work, we will focus our efforts to extend the use of this calibration rig platform for a large variety of sensors and sensor boards. In addition, the measured data stored by the sensor nodes during calibration are essential for further data mining and development of more effective calibration models.

In this first part, due to the dimension of the work, we focused on the design of an automated calibration rig with a proof of concept applied to calibration per se. We are aware of cross-sensitivity and temperature drift, however, we are planning to deliberately devote a separate investigation on this under Qatar environment (high temperature). Nevertheless, the designed system has proved capable of automatically and accurately mixing gases and hence able to study/quantify cross-sensitivity of specific gases to specific sensors along with temperature drift with the capability to change the chamber temperature from room temperature up to 60 °C, which covers worldwide room temperatures including Qatar.

## Figures and Tables

**Figure 1 sensors-20-02341-f001:**
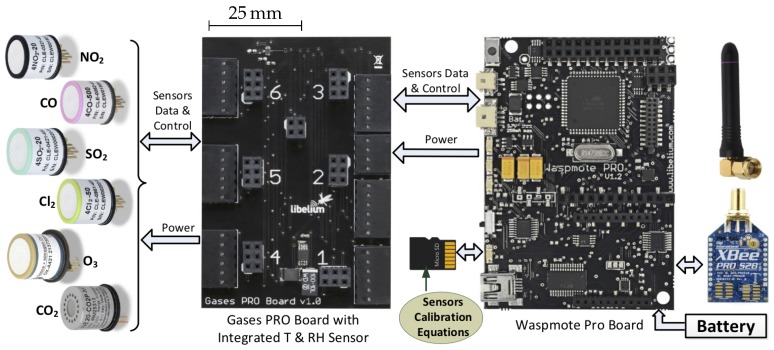
Sensor node for air-quality monitoring.

**Figure 2 sensors-20-02341-f002:**
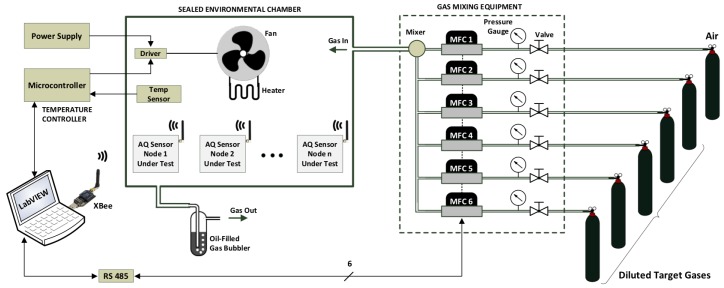
Architecture of the proposed calibration setup.

**Figure 3 sensors-20-02341-f003:**
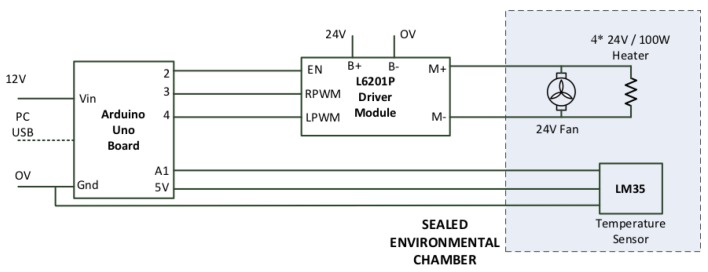
Schematic of the temperature control system.

**Figure 4 sensors-20-02341-f004:**
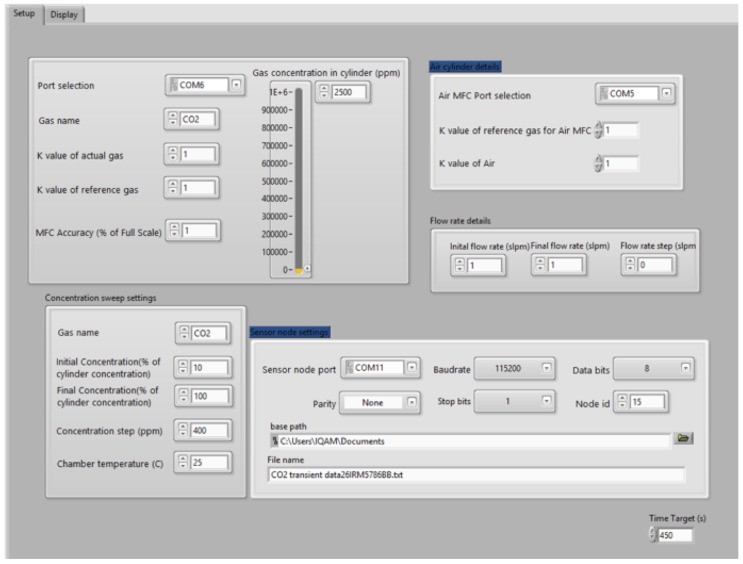
LabVIEW VI running the calibration rig during data collection phase.

**Figure 5 sensors-20-02341-f005:**
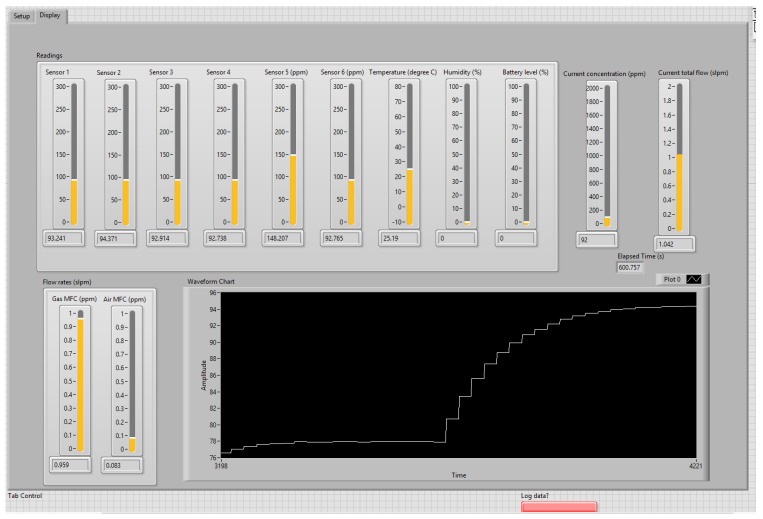
Graphical User Interface (GUI) displaying the measurements of CO gas for calibration.

**Figure 6 sensors-20-02341-f006:**
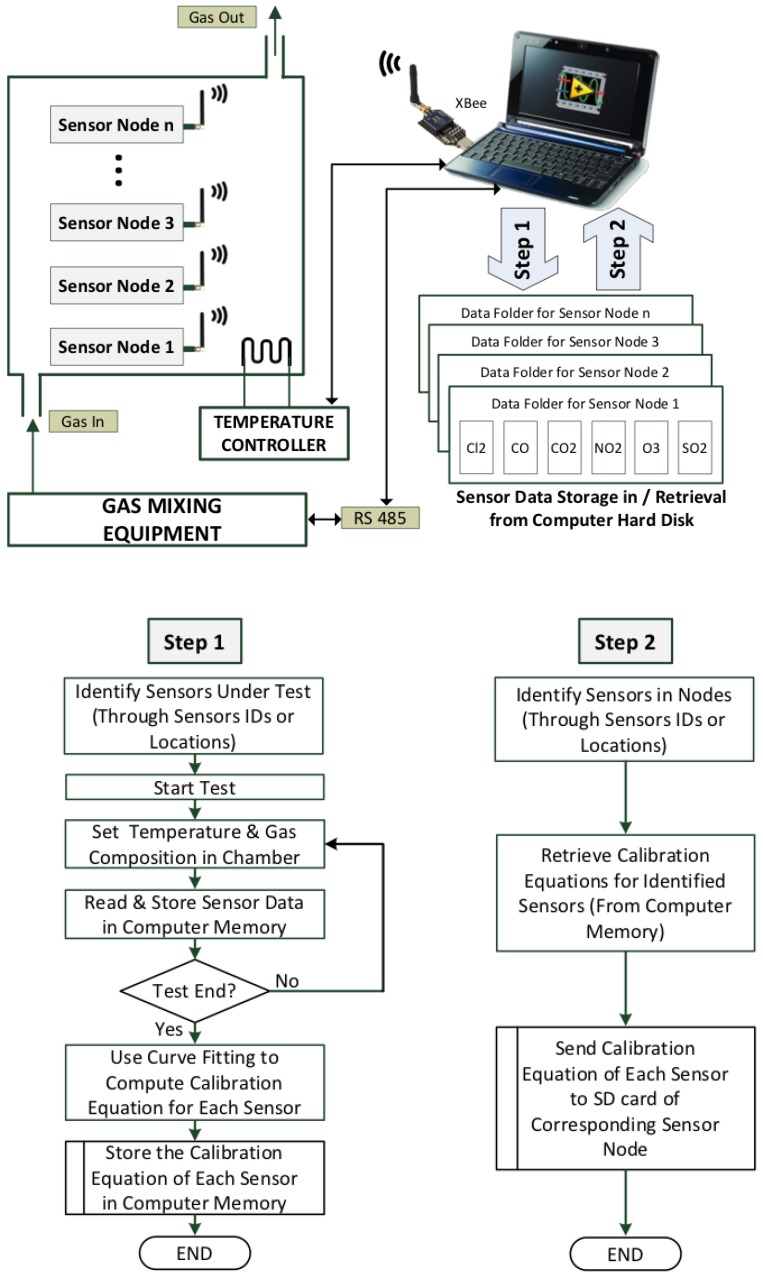
Sensor data collection and curve fitting (step 1); sensor calibration (step 2).

**Figure 7 sensors-20-02341-f007:**
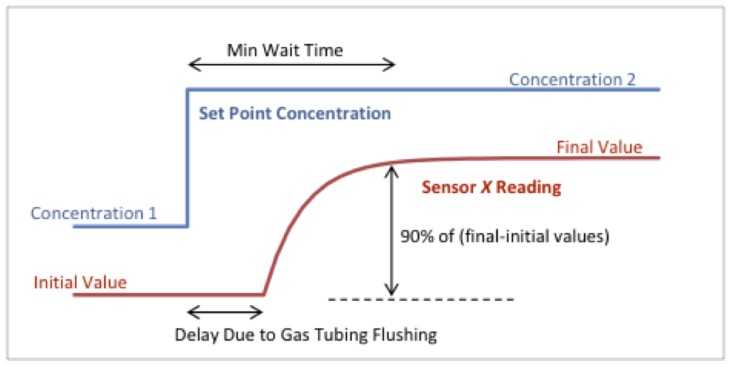
Sensor response following the set point concentration of the target gas sent to the mass flow controller (MFC).

**Figure 8 sensors-20-02341-f008:**
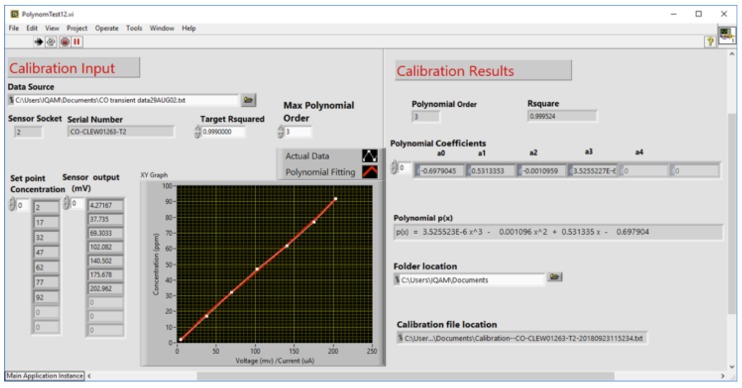
Graphical User Interface for the LabVIEW VI for the polynomial fitting.

**Figure 9 sensors-20-02341-f009:**
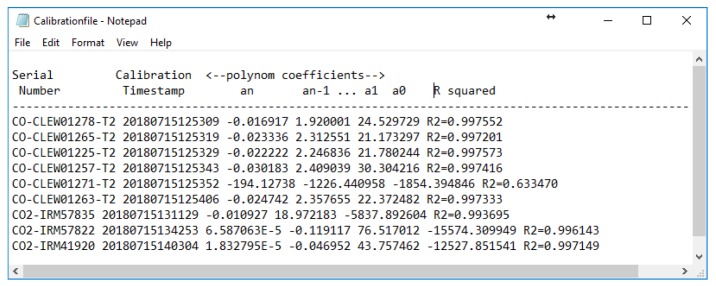
Polynomial coefficients stored in the SD card at the sensor node. One line for each gas sensor with its absolute identifier (serial number).

**Figure 10 sensors-20-02341-f010:**
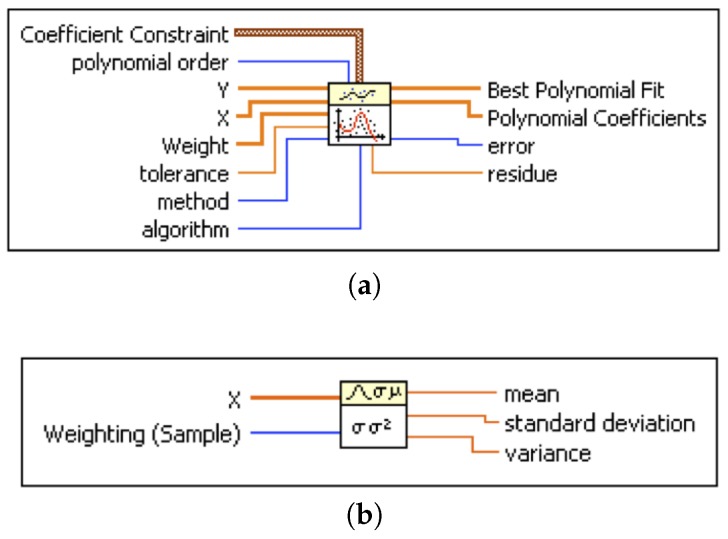
General Polynomial Fit VI (**a**) and Standard Deviation and Variance Virtual Instrument (**b**) of LabVIEW [48].

**Figure 11 sensors-20-02341-f011:**
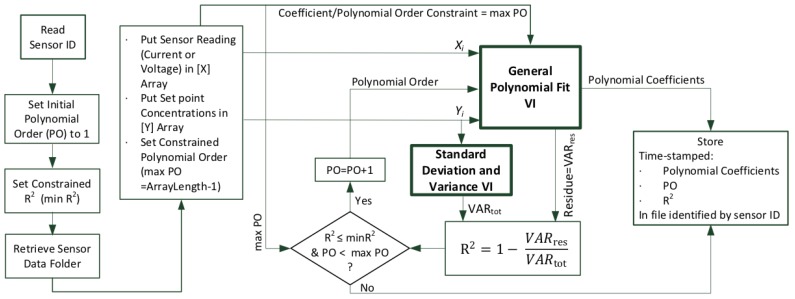
Implementation of the curve fitting for a single sensor using a dedicated LabVIEW VI.

**Figure 12 sensors-20-02341-f012:**
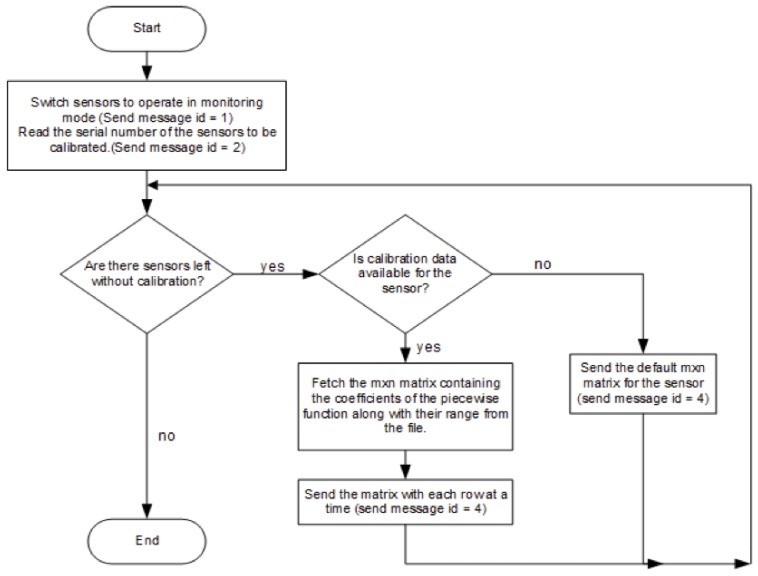
Process of uploading the polynomial coefficients to the sensor node.

**Figure 13 sensors-20-02341-f013:**
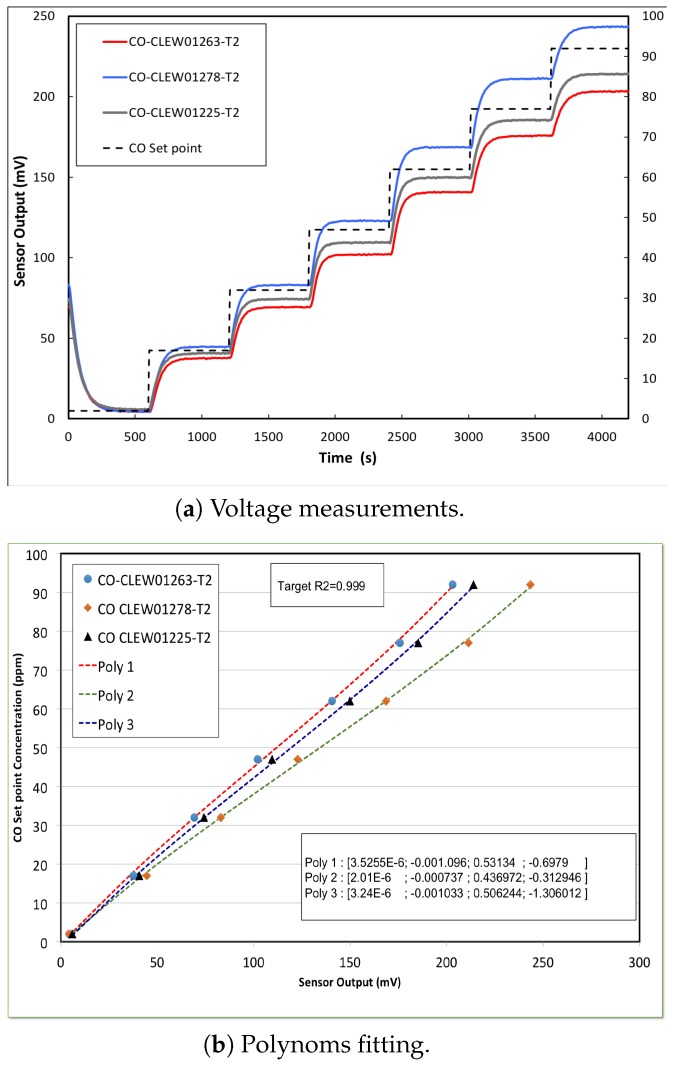
CO sensor response to stepping of CO concentration before calibration, (CO-CLEW01263-T2, R_2_ = 0.997333) (CO-CLEW01278-T2, R_2_ = 0.997552) (CO-CLEW01225-T2, R_2_ = 0.997573).

**Figure 14 sensors-20-02341-f014:**
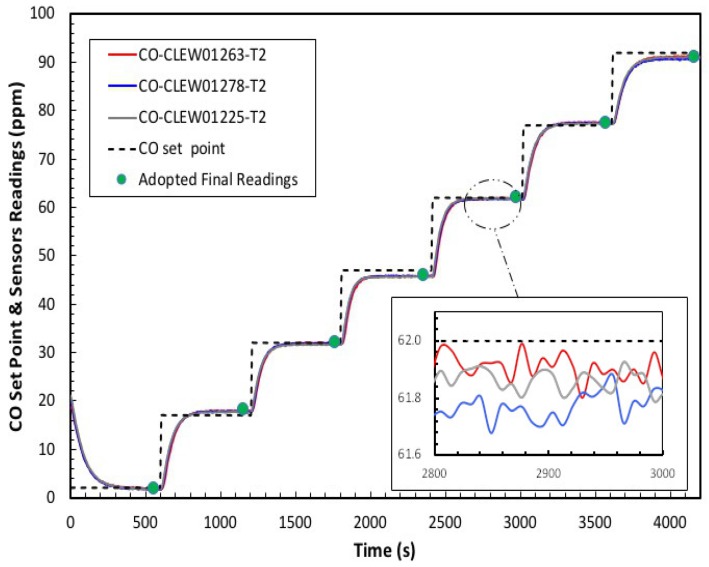
CO set points and sensors detected CO concentration after calibration (ppm).

**Figure 15 sensors-20-02341-f015:**
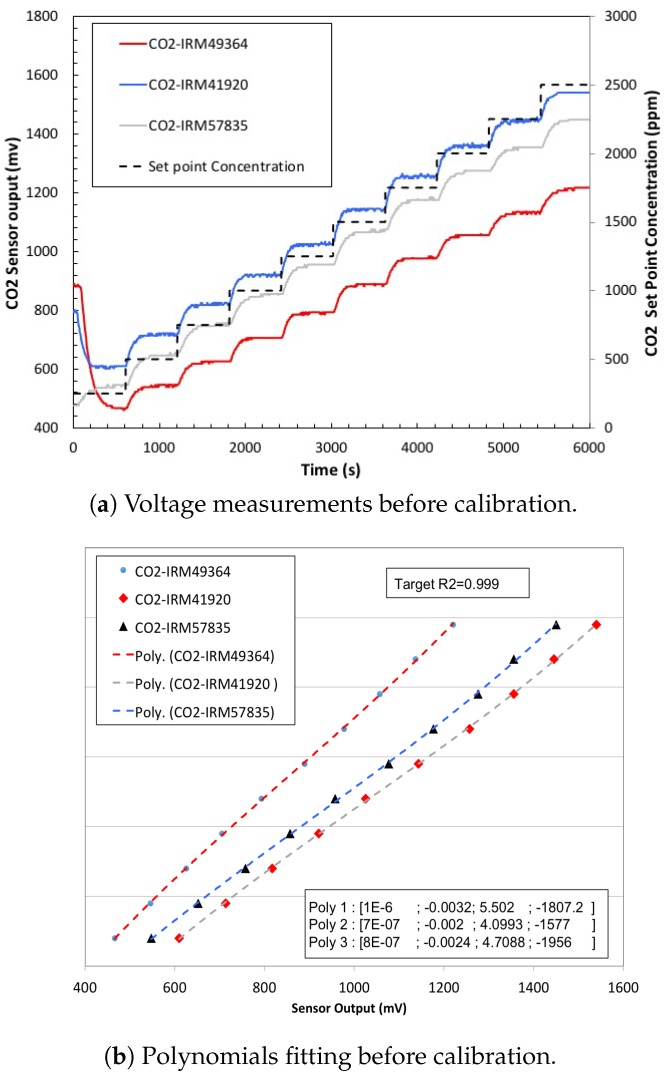
CO_2_ sensor response to stepping of CO_2_ concentration before calibration, (CO2-IRM49364, R_2_ = 0.999922) (CO2-IRM41920, R_2_ = 0.997149) (CO2-IRM57835, R_2_ = 0.993695).

**Figure 16 sensors-20-02341-f016:**
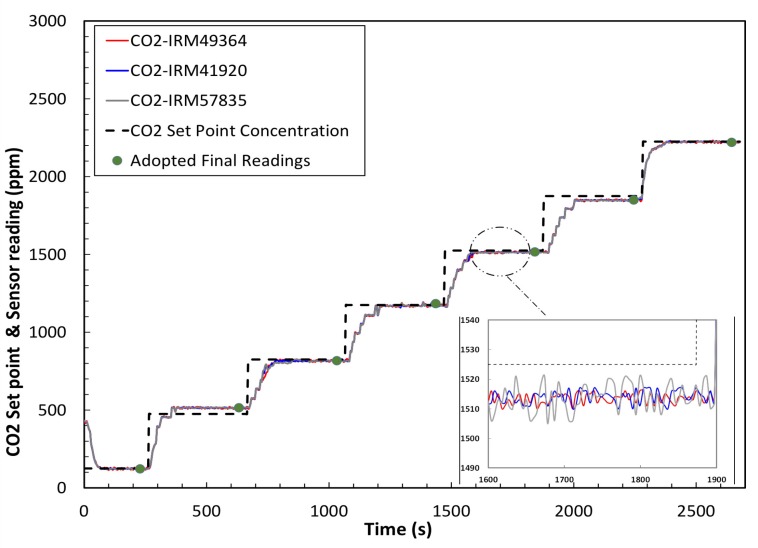
CO_2_ set points and sensors’ detected CO_2_ concentration after calibration (ppm).

**Table 1 sensors-20-02341-t001:** Specifications of the used sensors, where BME280 is Humidity—temperature and pressure sensor, T90 is the amount of time that it takes for the detector to measure 90% of the maximum concentration level, and T63 values indicate how long the measurement takes to reach 63%.

IAQ Parameter	Sensor Model	Nominal Range	Accuracy (ppm)	Response Time (s)	Sensor Type
Cl_2_	4-Cl_2_-50	0–50 (ppm)	±0.1	T90 ≤ 30	Electrochemical
CO	4-CO-500	0–1000 (ppm)	±1	T90 ≤ 30	Electrochemical
CO_2_	INE20-CO_2_P	0–5000 (ppm)	±50 (0–2500 ppm)	T90 ≤ 60	NDIR
-NCVSP	±200 (2500–5000 ppm)
NO_2_	4-NO_2_-20	0–20 (ppm)	±0.1	T90 ≤ 30	Electrochemical
O_3_	OX-A431	0–18 (ppm)	±0.2	T90 ≤ 45	Electrochemical
SO_2_	4-SO_2_-20	0–20 (ppm)	≤0.1	T90 ≤ 45	Electrochemical
T & RH	BME280	−40–85 °C (T)	±1 °C (T)	T63 ≤ 1.65 (T)	
		0–100% (RH)	±3% (RH)	T63 ≤ 1 (RH)

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
