# Peer review of "A Smart Rig for Calibration of Gas Sensor Nodes"

_sensors, 2020, doi:10.3390/s20082341_

Round 1
Reviewer 1 Report
General Comments:
The manuscript presents an interesting work, with a technological and applied approach. The novelty of the work is clearly presented: A dedicated calibration system for gas sensor nodes.
The manuscript addresses the well-known pre-calibration problem for gas sensors, which normally suffer from aging, drift or poisoning, as well as poor repeatability.
Authors should highlight the well-defined behavior of the calibration curves (Figures 14 ii and 16 ii). They are almost straight lines (what are the R-square for such fitting?). However, the authors did not make any repeatability evaluation. How reliable are the calibration curves fitted? How reliable are they for the next gas trial?
It is not correct to raise the order of the polynomial until it reaches a good fit for each trial. If you take the polynomial order equal to the number of points, you will get the perfect fit R-square = 1, but this is meaningless if there is no repeatability. Instead, authors should evaluate the dispersion of calibration curves for some gas trials repetitions.
Specific Comments:
Line 105: please correct “...after every re-calibration...”.
Lines 219 to 247: The explanation of polynomial fit is overly detailed.
The subVIs in Figure 11 belongs to the LabView library. Do not explain its use in the article, as this information can be found in the LabView manual.
Lines 247 to 253: Two comments: first, the authors do not need to define the well-known mean, variance, and R-square expressions. Second, it makes no sense to determine the mean for X or Y values, because they correspond to different set-point gas concentrations. (X and Y are the sequence of values in which the polynomial fitting is made).
Line 295 and 296: I suppose the correct is “Figure 16 shows the measurement results and the evaluation of the CO2 sensor before calibration, while Figure 17 shows them after calibration”.
Lines 282 to 298: The discussion is strange. The sensors responses in mV are different because the sensors are not identical. That is why we have raised the respective calibration curves. Then, different curves are not “incorrect” as said in line 284. Furthermore, figures 15 and 17 are meaningless because it is obvious that the respective fitted polynomials produce “identical responses”.
Reviewer 2 Report
In the manuscript entitled “A Smart Rig for Calibration of Gas Sensor Nodes”, the authors, M. Benammar et al, report a dedicated smart calibration rig with a set of novel features enabling simultaneous calibration of multiple sensors.
First of all, the abstract section should be reduced, reporting directly the motivation and the best obtained results; the first part from lines 1 to 7 should be deleted.
In the introduction, the authors should motivate the choise of the emplojed commercial gas sensors.
In my opinion, to optimize the device calibration, the sensing data should be compare with those of tha gas analyzers that are used, at the international level, for air gas monitoring. Please, discuss.
In the all manuscript the English stile should be improved.
I can accept with major revisions.

Reviewer 3 Report
An automated LabVIEW-controlled calibration rig is reported for the simultaneous calibration of multiple sensors retreated from the field) which offers pre-programmed calibration steps and requires minimal operator intervention. Details are given on the used sensor nodes, hardware and software operation of the calibration rig, and exemplary results of calibration to CO and CO2 in a sealed environmental chamber. The paper is concise and will find interest in the field. However, before publication it deserves minor improvement according to the points listed below.
- Authors should briefly describe design and function of the electrochemical gas sensors used in this study with emphasis on parameters (drift, ageing, temperature dependence) causing need for (periodical) calibration.
- There are various other types of low-cost gas sensor principles beyond electrochemical sensors, e.g. such based on nanomaterials (see: -Advanced Nanomaterials for Inexpensive Gas Sensors – Synthesis, Integration and Applications, Eduard Llobet (ed), Elsevier 2020); -Xu et al., “Piezoresistive Microcantilevers 3D-patterned using ZnO-Nanorods@Silicon-Nanopillars for Room-Temperature Ethanol Detection”, 20th Intern. Conf. Solid-State Sensors, Actuators and Microsystems (Transducers 2019 - EUROSENSORS XXXIII), Berlin, 23 - 27 June 2019, pp. 1211-1214; DOI: 10.1109/TRANSDUCERS.2019.8808821; -Markiewicz et al., “Ultra Low Power Mass-Producible Gas Sensor based on Efficient Self-Heated GaN Nanorods”, 20th Intern. Conf. Solid-State Sensors, Actuators and Microsystems (Transducers 2019 - EUROSENSORS XXXIII), Berlin, 23 - 27 June 2019, pp. 1321-1324; DOI: 10.1109/TRANSDUCERS.2019.8808234). Authors should refer to these paper as examples for other types of low-cost gas sensors which presumably require periodical calibration, e.g., using the procedure described here.
- Table 1: give “sensor type” of the BME280 and indicate the definition of “T90” and “T63”.
- Line 45: give prizes of “low-cost gas sensors”, e.g., of the used electrochemical ones.
- In Fig. 1: include a scale bar.
- Line 83: “O3, NO2 and SO2”, numbers should be given as indices. Applies also to “CO2” in line 91 and “I2C” in line 97. Authors should check their manuscript accordingly.
- Line 98: “Libellium” -> “Libelium”.
- The photograph of the Gas Mixing Equipment in Fig. 3 is not very informative. Which components are shown there? What is the size size of the setup?
- Line 214: “id” -> “ID”? Is it related to the parameter “message id” in Fig. 13?
- The graphical user interfaces in Figs. 5, 8, 9 may not be necessary to understand the operation principle of the developed calibration procedure and may be skipped. The same applies to the SD-card-stored value list shown in Fig. 10.
- The fitting procedure described on page 10 is not clear and should be revised. What is the difference between Eqs. (2), (3), and (4)? The parameter “fi” in Eq. (2) is not defined and the square sign to the term in brackets is missing? The parameter “w” in line 249 is not defined.
- The abbreviation list at the end of the paper may not be complete, e.g. “AT” in line 159, “SD”, “ID”, “EEPROM”, etc. elsewhere. There are further abbreviations defined in the text, which should also be added to the list at the end.
- Use larger font size of labels especially in Figs. 6,11, 12, 13.
- There is no reference to Fig. 16 in the text.
- In Figs. 15(i) and 17 areas are highlighted by an ellipse but not explained.
- The method for determining the dependence of the sensor output on the set point concentration appears not to be straightforward. Authors should derive a physical-based model available to describe the sensitivity of electrochemical sensors, which at least might limit the order of the used polynomial?
- What is the total time required for a typical calibration procedure? After what time a used sensors have to be calibrated again?
Reviewer 4 Report
This paper presents a computer controlled measurement rig for calibrating gas sensors mounted on wireless sensing boards. The architecture of the board and the sensors used are commercially available. Therefore, the main originality within the paper lies on the calibration approach envisaged and implemented.
The gas mixing system comprises a few mass-flow controllers (MFCs), which enables the mixture of gases with a balance gas (dry air) employing calibrated gas bottles. The gas mixture is delivered to a chamber in which ambient temperature can be controlled and where gas sensor boards are placed for calibration measurements.
The approach of using a gas mixture and delivery system using MFCs and calibrated gas cylinders is conventional and employed almost universally in gas sensor research and sensor manufacturing industry.
It is surprising that sensors are calibrated for environmental applications, but the system does not comprise a humidity generation/control system so sensors can be tested and calibrated under conditions that approach real scenarios.
Important information is missing: For example, the volume of the chamber, the total gas flow, the range of temperatures achievable inside the chamber and the uncertainty associated to temperature. With this it is difficult to assess what is the time needed to change concentrations between consecutive measurements. In addition, the range of achievable concentrations is not given.
The experimental calibration is demonstrated employing electrochemical sensors. This calibration is illustrated for CO. However, electrochemical sensors are not 100% selective. It is surprising that a calibration is demonstrated in the presence of a single gas if the system, in principle, seems suited for delivering multicomponent gases. Once more, running a calibration to a target gas in the presence of a few interfering species would be closer to the real application envisaged.
The calibration consists of determining the coefficients of a polynomial fitting for sensor response. What is the reason for using such an approach? How is the degree of the polynomial fitting to be used set?
Considering my comments above, I cannot recommend the publication of this paper in Sensors.
Round 2
Reviewer 1 Report
General comments:
In my opinion the work has a serious methodological problem: As I pointed in the first review, it is not correct to raise the order of the polynomial until it reaches a good fit for each trial. This empirical methodology may be acceptable if the authors could demonstrate a high repeatability of the sensor response (how repeatable are the calibration curves of the sensors?).
Specific comments:
Lines 44 to 46: Just say that the price of the sensors is around tens of USD.
Line 48: I suppose the correct is “GaN Nanorods”.
Figure 3 (Gas Mixing Equipment) is not necessary. Instead, the authors could replace it with a photo from the gas test chamber.
I think the introduced item “3.3. Modelization of the Calibration Drift” should be removed. That item discuss the dynamic profile of the sensors response, but the calibration curves are obtained from stable data on concentration steps. That item is an unnecessary complication. In summary, the authors choose fit the stable data points by a polynomial of order 2 or 3. It is OK. It is sufficient.
Lines 311 to 313: The sensor responses are not “incorrect”. It is better to say: “Although the sensors have the same specifications, they are not identical. Thus, the response voltages produce different calibration curves, as shown in Figure 14(ii).”
Figure 15 (ii) is wrong because: 1) The sensor output is not “concentration (ppm)”, but “voltage (mV)”, and 2) after calibration, the expected ppm vs. ppm curve should be a perfect 1:1 straight line. This is the reason that kind of plot is unnecessary. The calibration curves are the “set point concentration (ppm)” vs. “sensor output voltage (mV)” for each sensor (Figure 14(ii)). The same discussion is valid for figures 16 and 17. Then, I suggest remove Figures 15 and 17.
Author Response
Dear Reviewer 1,
Thank you very much for your support,
the revision is in the attached file.

Reviewer 2 Report
Now, in my opinion, the revised version of the manuscript is acceptable for publication.
Author Response
Dear Reviewer 2,
Thank you very much for your valuable comments serving to improve this paper.
Reviewer 4 Report
Authors have revised their manuscript substantially and some improvements are clearly visible in this amended version. However, there are still a few aspects that need further revision and clarification.
- Humidity effects. If the calibration system is to apply generally to the broad spectrum of commercially available sensors that can be used in environmental monitoring, humidity generation and testing is a must. Also electrochem and many other sensors can be perfectly usable in a wide range of moisture levels, but this does not mean that their response is not seriously affected by changes in the humidity background.
- In the amended paper new information about the calibration of response dynamics is given. This is judged interesting but I have a concern. Despite the fact that the gas flow is high 1 L/min, the time needed for changing the concentration inside the test chamber is 450 seconds (more than 7 minutes), as stated by authors. This is probably due to the fact that the volume of the test chamber is very high. Under these conditions, the dynamic sensor response observed may be completely dominated by chamber dynamics rather than by the dynamics of the detection mechanism. In fact sensor response dynamics for commercially available electrochems or semiconductor sensors are in the tens of seconds. Authors should explain better what exactly is the interest in modelling something that heavily depends on their particular set-up. Is this going to help expediting the calibration process?
Author Response
Dear Reviewer 4
Thanks you very much for your valuable revision,
you find in the attached file my response to each comment.
